# Impact of the COVID-19 Pandemic on Total and Cause-Specific Mortality in Pavia, Northern Italy

**DOI:** 10.3390/ijerph19116498

**Published:** 2022-05-26

**Authors:** Pietro Perotti, Paola Bertuccio, Stefano Cacitti, Silvia Deandrea, Lorenza Boschetti, Simona Dalle Carbonare, Stefano Marguati, Simona Migliazza, Eleonora Porzio, Simona Riboli, Ennio Cadum, Lorella Cecconami, Anna Odone

**Affiliations:** 1Health Protection Agency of Pavia (ATS Pavia), 27100 Pavia, Italy; pietro_perotti@ats-pavia.it (P.P.); silvia_deandrea@ats-pavia.it (S.D.); lorenza_boschetti@ats-pavia.it (L.B.); simona_dalle_carbonare@ats-pavia.it (S.D.C.); stefano_marguati@ats-pavia.it (S.M.); simona_migliazza@ats-pavia.it (S.M.); eleonora_porzio@ats-pavia.it (E.P.); simona_riboli@ats-pavia.it (S.R.); ennio_cadum@ats-pavia.it (E.C.); lorella_cecconami@ats-pavia.it (L.C.); 2Department of Public Health, Experimental and Forensic Medicine, University of Pavia, 27100 Pavia, Italy; stefano.cacitti01@universitadipavia.it (S.C.); anna.odone@unipv.it (A.O.)

**Keywords:** mortality, cause of death, pandemic, COVID-19

## Abstract

The COVID-19 pandemic had an unfavorable impact on overall mortality in Italy, with the strongest consequences in northern Italy. Scant data are available on cause-specific mortality. This study aims at investigating the impact of the pandemic on the overall and cause-specific mortality in one province in northern Italy, Pavia. We linked individual-level administrative data (i.e., death certification and population data) from the Health Protection Agency (HPA) in Pavia province between 2015 and 2020. We computed age-standardized mortality rates (Italian population 2011) by cause, sex, and calendar year, and computed the rate ratio and the corresponding 95% confidence intervals to compare rates in 2020 versus 2015–2019. The 2020 excess total mortality in Pavia was 24% in men and 25% in women, reaching rates of 1272.6/100,000 and 1304.4/100,000, respectively. Significant excesses were found for infectious and parasitic diseases, excluding COVID-19 (about +30% in both sexes); respiratory system diseases (44% in men; 30% in women); and dementia and Alzheimer’s disease among men (24%). Reductions were reported for neoplasms (−14% in men); cerebrovascular diseases (−25% in men); and ischemic diseases (−13% in women), but also for transport accidents in men. COVID-19 was the third cause of death in both sexes with rates of 274.9/100,000 men (859 total deaths) and 213.9/100,000 women (758 total deaths). Excess mortality in Pavia was higher than Italy but lower than Lombardy. Increases in mortality from causes related to infectious and respiratory diseases can likely be explained by underdiagnosed deaths from COVID-19.

## 1. Introduction

Of the European countries, Italy was the first and most heavily-hit by the coronavirus disease 2019 (COVID-19) pandemic—specifically Lombardy in the northern region [1,2]. Over 2 million COVID-19 cases (2,169,503) with 77,165 deaths were identified up to the end of December 2020 in Italy; 471,311 cases (21.7% of the total) with 25,362 fatalities (32.9%) were in Lombardy and most of them occurred during the first wave [3].

The pandemic’s impact on healthcare delivery has gone further than simply the number of deaths from COVID-19; it has also caused many other victims due to the reduced capability of the system to provide care for health issues that are not directly related to the virus [4,5,6]. Therefore, excess mortality during the COVID-19 pandemic is the most reliable indicator of its burden [7]. Mortality data provide a complete assessment and contribute to the estimate of the direct and indirect COVID-19 pandemic burden. Estimates of excess mortality from all causes are heterogeneous across the country with dramatic increases in northern geographical areas, but there remain no figures of net mortality in central and southern Italy. Figures are periodically updated by the Italian National Institute of Statistics (ISTAT) and the National Institute of Health (ISS), and several studies estimated the excess total mortality in Italy [8,9,10,11]. However, a few studies have been published on mortality data by specific cause of death [12,13,14]. Analyses of cause-specific excess mortality are important to understand and quantify the determinants and mediators of the short and long-term impact of the COVID-19 pandemic on societal health and wellbeing.

The current analysis focused on cause-specific excess mortality in the Pavian urban and rural areas in northern Italy (Lombardy region), with around 540,000 inhabitants. Based on internal data registered at the Epidemiological Observatory in the Health Protection Agency (HPA) of Pavia, the province, with 186 municipalities, has the highest regional old-age index (197.8 versus 162.2 in Lombardy and 168.9 in Italy), and a high overall mortality (age-standardized mortality rate of 92.9/100,000 versus 81.6 in Lombardy and 86.7 in Italy). From the beginning of the pandemic to the end of December 2020, 25,263 residents in Pavia province were diagnosed with COVID-19 infection and 1617 consequently died [15]. The aim of this study therefore is to describe and compare the overall and cause-specific mortality before and during the COVID-19 pandemic in the Pavia province, separated by sex and using 2015–2019 as a reference period.

## 2. Materials and Methods

We linked individual-level administrative data from the HPA of Pavia. Detailed copies of the death certificates of Pavia residents were transmitted to the HPA for coding of the causes of death, according to the Tenth Revision of the International Classification of Diseases (ICD-10). Deaths that were due to COVID-19 were coded in accordance with the latest recommendations by the WHO [16]. Data entry in the mortality database was performed by specialized operators: 90% of records were automatically coded through an ad-hoc developed software used in the Lombardy region, and the remaining records (mostly from external cases, childhood mortality and other peculiar causes) were manually coded.

We computed age-standardized mortality rates (direct method) using as standard the Italian population (Census 2011) [17], for each cause of death (CoD), sex, and the calendar year between 2015 and 2020. In addition, age-standardized mortality rates for three age groups (i.e., 50–64, 65–79, ≥80 years) were computed for the leading CoD (i.e., neoplasms and cardiovascular diseases, plus COVID-19 disease). To compare the rate in the five-year period of 2015–2019 versus that registered in 2020, we computed the rate ratio and the corresponding 95% confidence interval (CI).

## 3. Results

Table 1 and Table 2 show the number of deaths and age-standardized mortality rates (ASR) from all causes and groups of causes in men and women, respectively, as well as the rate ratio between 2020 and the 2015–2019 period and the corresponding 95% CI. In Pavia province in 2020 we registered a total number of 859 deaths due to the COVID-19 among men and 758 among women, with an ASR of 274.9/100,000 and 213.9/100,000, respectively.

In the male population (Table 1), all-cause ASR in 2020 was 1272.6/100,000 inhabitants; this was higher than the previous period with a rate ratio of 1.24 (95% CI: 1.20–1.28). Considering the CoD in detail, the highest ASR was reported for neoplasms (312.1/100,000); followed by cardiovascular diseases (282.1/100,000); and COVID-19 (274.9/100,000). Significant increases in rates were observed for other CoDs, such as mental and behavioral disorders (rate ratio 1.50; 95% CI: 1.22–1.84), specifically for dementia and Alzheimer, but also diseases of respiratory system (1.44; 95% CI: 1.28–1.61), specifically pneumonia, and diseases of the musculoskeletal system and connective tissue (2.37; 95% CI: 1.06–5.30), though this last cause was based on a very few numbers.

Contrastingly, we observed a decrease in male mortality rates for the two major CoD, i.e., neoplasms (rate ratio 0.86; 95% CI: 0.80–0.92) and cardiovascular diseases (rate ratio 0.89; 95% CI: 0.83–0.96). These rates specifically decreased for cerebrovascular diseases, but mortality from transport accidents also showed a significant decrease (rate ratio 0.43; 95% CI: 0.23–0.80).

In the female population (Table 2), all-cause ASR in 2020 was 1304.4/100,000 inhabitants; this was higher than the previous period with a rate ratio of 1.25 (95% CI: 1.21–1.30). Among the studied CoD, cardiovascular diseases resulted as the first leading CoD (365/100,000), followed by neoplasms (263.5/100,000) and COVID-19 disease (213.9/100,000). An increase in mortality rates between 2020 and 2015–2019 was observed for diseases of the respiratory system (rate ratio 1.30; 95% CI: 1.16–1.47), mainly for flu virus, symptoms, signs, and abnormal clinical and laboratory findings that were not elsewhere classified (1.60; 95% CI: 1.35–1.90), as well as for external causes of mortality (1.21; 95% CI: 1.01–1.46). Diseases of the musculoskeletal system and connective tissue (1.70; 95% CI: 1.10–2.64), and diseases of the skin and subcutaneous tissue (2.22; 95% CI: 1.01–4.88) showed a significant increase in the female population, even if with a small number of observed deaths. On the other hand, none of the leading CoD that were considered in our analysis showed a significant reduction in mortality rate ratio if compared to the 2015–2019 span in female population, with a small and not statistically significant decrease in both neoplasms and cardiovascular diseases.

Table 3 shows the ASR and rate ratio between the calendar year 2020 and the five-year period of 2015–2019 for the three most important leading CoD, according to sex and age groups (i.e., 50–64, 65–79, 80 and over). Considering the COVID-19 disease, the highest ASR was observed in the oldest age group (≥80 years) with an ASR of 3137.3/100,000 in men and 1808.3/100,000 women. Mortality from neoplasms decreased in men in the age group 50–64 (rate ratio 0.77; 95% CI: 0.65–0.92) and 64–79 years old (rate ratio 0.85; 95% CI: 0.77–0.94). The oldest male group and any of the female age groups did not experience significant variations. Mortality rates from cardiovascular diseases showed very high values in the ≥80 years old groups for both sexes (4224.4/100,000 men, 3673.8/100,000 women), especially when compared to the 64–79 years old groups (604.0/100,000 men, 300.7/100,000 women). On the other hand, mortality rate ratios showed significant variations only in the ≥80 years old groups in the male population (rate ratio 0.83; 95% CI: 0.76–0.91).

Figure 1 shows the overall number of deaths from all-cause mortality in 2020 and the average deaths in the 2015–2019 period per month. Deaths were particularly high in the two months of March (1427) and April (1290), followed by December and November, while the lowest numbers of fatalities were observed during the summer months. Fatalities increased greatly in March and April 2020 (+149%) with a smaller increase in May and a second slighter peak from October to December. On the other hand, before the COVID-19 outbreak (January–February 2020) mortality rates were lower than those recorded in the previous five-year period (−5.5%) and no significant mortality excess emerged during the summer period (June–September 2020).

## 4. Discussion

Findings from this report, based on administrative data from the HPA of Pavia province in northern Italy, provide evidence of the huge impact of the pandemic on total mortality and some cause-specific figures. In this area, an excess in total mortality of 24% in men and 25% in women was found in 2020 as compared to that in 2015–2019. Interestingly, excesses were registered for infectious and parasitic diseases other than COVID-19 (about 30% in both sexes), diseases of the respiratory system (44% in men and 30% in women), and also dementia and Alzheimer’s disease in men (24%). Conversely, decreases were found in mortality from neoplasms in men (−14%), cardiovascular diseases, specifically cerebrovascular diseases in men (−25%) and ischemic diseases in women (−13%). In addition, a substantial decrease was reported for mortality from transport accidents in men (−57%), whereas a 20% excess mortality was reported for external causes in women. Figures in the older age groups showed decreases in neoplasms by 23% in the male population aged 50–64 years and by 15% in males aged 65–79 years, as well as decreases in cardiovascular diseases in the oldest males (≥80 years old) by 17%.

In 2020, the first two leading CoD were neoplasms, followed by cardiovascular diseases in men, while cardiovascular diseases were followed by neoplasms in women. COVID-19 alone was the third most common leading cause of deaths in Pavia province in both sexes with an ASR of 274.9/100,000 in men and 213.9/100,000 in women.

Several studies estimated excess mortality in Italy by using different approaches. Some of them compared the number of deaths in 2020 and the average registered in the previous five-year period [3]. A study that was based on official national data and used a statistical model to consider demographic changes and temporal improvements in mortality estimated an excess mortality of more than 90,000 deaths in 2020 [11], which was lower as compared to the excess of 100,526 deaths obtained by the comparison with the average number of deaths in 2015–2019 [3].

Lombardy was the most affected Italian region with a total of 136,249 deaths in 2020 and a mortality excess of 36.6%, although COVID-19 impact on this excess was 68.8%, which was lower than the national average [3]. The Pavia province mortality excess of 2020 was therefore higher than the Italian one, but lower if compared to the Lombardy mean [14]. On the other hand, the impact of COVID-19 infection on excess mortality was 75.3%, higher than both regional and national data. We may assume that at least a part of these huge differences in the proportions of mortality excess that were attributed to COVID-19 infection could be explained by a sub-diagnosis in some of the Lombardy provinces that were more heavily involved during the first wave, with many patients dying without being tested for infection [1,18].

Regarding the excess-of-mortality trend during the pandemic period, our data are consistent with regional data. During the first pandemic phase (March–May 2020), the excess of mortality in the province of Pavia was 107%, similar to Lombardy (111.8%) and much higher than the national data (31.7%) with most of the total mortality excess concentrated in northern regions [19]. During the second wave of the pandemic (October–December 2020), excess mortality in the province of Pavia was 27.1%, lower than both regional (37.1%) and national (32.3%) excess. In this phase, the distribution of excess mortality was uniform through the whole country, and many regions of central and southern Italy, but even some northern regions, suffered an increase in mortality that was not registered during the first wave [12,20]. To our knowledge, very few reports have been published that analyzed excess mortality from specific causes of death that were not COVID-19 during 2020. Figures based on data from the municipality of Rome found a 10% all-causes deaths excess in 2020 as compared to the 2015–2018 period. This was lower than the national mean with the highest increase during the second pandemic wave (October–December 2020) and in the oldest population (+18% in the ≥80 years old group). Regarding cause-specific mortality, in accordance with our results, a decrease in mortality that was caused by cardiovascular diseases and neoplasms was also reported [12]. Reductions in mortality from neoplasms emerged from national figures (−4.3%) during the first pandemic wave but were less evident from the data on Lombardy (−0.8%) [9].

The pandemic’s impact on the health-care system therefore, does not seem to have negatively influenced the management or treatment of cardiovascular diseases and neoplasms. More likely, if there was a negative impact it was easily covered by the reduction in deaths that were attributed to these causes because of the harvesting effect. More vulnerable subjects who could have died during 2020 because of cardio circulatory diseases or neoplasms probably deceased during the first wave of the pandemic, leading to the reduction in mortality that was attributed to these two most common causes [21]. Nevertheless, we may assume that the paralysis of screening services due to the pandemic [22] and peoples’ possible unhealthy behaviors during the lockdown period, as well as infection consequences on the cardiovascular system [23], could represent an important risk factor for an increased burden of these diseases in the near future.

Increased mortality from infectious and parasitic diseases other than COVID-19 that was observed in Pavia is consistent with other previous studies based on Lombardy data during the first pandemic wave, despite an overall decrease at national level [9]. The COVID-19 pandemic is probably associated with many of the fatalities that were classified as due to respiratory diseases, among which pneumonia was the most common cause of death. Patients with respiratory symptoms may not have been diagnosed as infected by the virus due to a lack of testing availability, or they may not have received the proper treatment for their medical conditions in the situation of emergency because of a lack of resources [9,24]. The lack of medical personnel and resources [25] (but probably even health policy-makers’ suggestions to minimize the risk of infection among sanitary operators) is arguably the main cause of the increased mortality that was attributed to symptoms, signs, abnormal findings, and poorly defined causes, even if this only significantly affected women. Increases in mortality from dementia and Alzheimer’s disease have previously been reported in Lombardy, but also in other areas on an international level [9,26,27]. Investigations to specifically assess the possible psychological distress due to the pandemic (and possibly due to lockdown), as well as details regarding the significantly different results between the sexes are needed.

Concerning external causes of mortality, other studies reported a decrease in suicides [28] and car accidents during the first pandemic period [29], ahead of an increase in domestic accidents [30]. These findings are expected since the emergency restrictions caused consequent drastic reductions in the use of all vehicles. Many surveys have highlighted that the pandemic dramatically affected mental health [31]; thus, trends in suicide should be monitored globally, both within countries and across communities [32].

Data on cause-specific mortality are of vast importance for public health, to understand the true changes in mortality as well as the magnitude of disease burden during the pandemic that was not directly due to SARS-CoV-2 infection. Estimates on overall excess mortality were reported for several countries worldwide [7]; however, data on CoD were more difficult to retrieve. Partially consistent with ours, findings from a study in the United States showed an increase of about 16% in the number of deaths in 2020, with increases in mortality from heart diseases (+4.8%); unintentional injury (11.1%); Alzheimer’s disease (9.8%); and diabetes (15.4%), while deaths that were caused by cancers did not show significant variation [27]. Another study from Brazil found decreases in mortality from both cancer and cardiovascular diseases [33]. These data are somehow consistent to our findings, even though they need to be contextualized and could be in part explained by the harvesting effect. In the interpretation of our data we should consider that demographic changes and the secular reduction in mortality rates may affect the excess mortality estimates. On the other hand, we used complete and official data regarding all the residents of one of Italy’s provinces in the country’s most heavily affected region, accounting for more than 500,000 inhabitants. In fact, this study allowed us to deeply investigate and quantify the COVID-19 pandemic impact on the overall and cause-specific mortality in Pavia province, comparing the figures from 2020 with those in the period 2015–2019.

Future studies are necessary to investigate both the short and long-term effects of the pandemic on mortality. Surveillance measures should be implemented, especially regarding causes of death, such as cancer and cardiovascular diseases, that show an apparent downward trend during emergency periods [34]. Delays in diagnosis due to the suspension of screening services, as well as infection consequences on the cardiovascular system could cause a return of these diseases’ impact on overall mortality.

## 5. Conclusions

This descriptive study provides evidence of excess mortality in Pavia in northern Italy (Lombardy region), specifically for infectious and parasitic diseases, excluding COVID-19; diseases of the respiratory system; and dementia and Alzheimer’s disease among men. Although the first two leading causes of death in 2020 were neoplasms and cardiovascular diseases, some reductions in their mortality emerged. The COVID-19 disease was the third cause of death. Our findings are partially consistent with previous reports; in particular, the 2020 excess mortality in Pavia was higher than Italy but lower than Lombardy. Finally, increases in mortality from causes related to infectious and respiratory diseases are likely due to an under-diagnosis of certified COVID-19 deaths.

## Figures and Tables

**Figure 1 ijerph-19-06498-f001:**
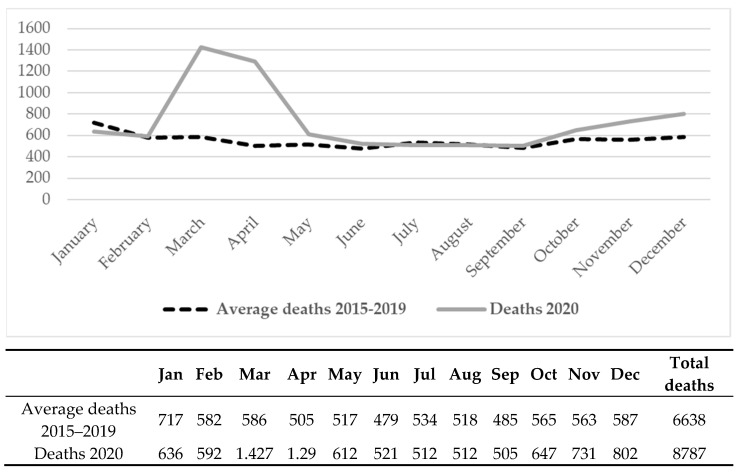
Per-month distribution of mortality events in 2020 and average number of deaths per-month in 2015–2019 period. Province of Pavia, Italy.

**Table 1 ijerph-19-06498-t001:** Number of deaths and age-standardized ^1^ mortality rates (ASR) from all causes and groups of causes in men at all ages in 2015–2019 and 2020, the rate ratio and the corresponding 95% confidence interval (CI). Province of Pavia, Italy.

	2015–2019		2020		2020 vs. 2015–2019		
	Average Deaths	ASR	Deaths	ASR	Rate Ratio	95% CI	
Total, all causes of death	3047.2	1027	4016	1272.6	**1.24**	**1.20**	**1.28**
**Group of causes**							
COVID-19	-	-	859	274.9	-	-	-
Infectious and parasitic diseases (COVID-19 excluded)	88	29.7	121	38.5	**1.30**	**1.06**	**1.59**
Neoplasms	1055.6	363.3	963	312.1	**0.86**	**0.80**	**0.92**
Malignant neoplasms of lymphoid, haematopoietic and related tissue	12.4	4.2	19	6.0	1.44	0.86	2.41
Endocrinuous, nutritional and metabolic diseases	101.8	34.7	119	37.8	1.09	0.89	1.33
Diabetes mellitus	80	27.3	88	28.1	1.03	0.82	1.30
Mental and behavioral disorders	74.4	24.1	121	36.1	**1.50**	**1.22**	**1.84**
Diseases of the nervous system and sense organs	129	43.3	125	39.9	0.92	0.76	1.12
Dementia and Alzheimer’s	111.6	36.2	151	45.0	**1.24**	**1.04**	**1.49**
Cardiovascular diseases	951.6	315.7	913	282.1	**0.89**	**0.83**	**0.96**
Ischemic heart diseases	344	115.2	332	105.4	0.91	0.81	1.03
Cerebrovascular diseases	225.4	74.2	185	55.9	**0.75**	**0.64**	**0.88**
Diseases of respiratory system	252.6	82.6	384	118.9	**1.44**	**1.28**	**1.61**
Flu virus	3.6	1.2	9	2.6	2.20	0.98	4.92
Pneumonia	93.4	30.6	156	48.5	**1.59**	**1.32**	**1.90**
Chronic lower respiratory tract diseases	97	31.5	119	37.0	1.17	0.96	1.44
Diseases of the digestive system	113.6	38.3	109	35.5	0.93	0.75	1.14
Diseases of the skin and subcutaneous tissue	3	1	4	1.2	1.14	0.38	3.45
Diseases of the musculoskeletal system and connective tissue	3.6	1.2	9	2.9	**2.37**	**1.06**	**5.30**
Diseases of the genitourinary system	73	23.8	65	19.3	0.81	0.62	1.06
Complications of pregnancy, childbirth and the puerperium	0	-	0	-	-	-	-
Certain conditions originating in the perinatal period	2.6	1.1	6	2.9	2.62	0.99	6.90
Congenital malformations, deformations and chromosomal abnormalities	3.8	1.4	5	1.7	1.23	0.46	3.32
Symptoms, signs and abnormal clinical and laboratory findings, not elsewhere classified	49.4	16.6	65	20.2	1.22	0.93	1.61
External causes of mortality	132.8	46.1	129	42.5	0.92	0.76	1.12
Transport accidents	25.2	9.2	11	4.0	**0.43**	**0.23**	**0.80**
Accidental falls	28.8	9.4	39	12.3	1.30	0.91	1.85
Suicide and intentional self-harm	33.2	12.0	32	10.9	0.91	0.62	1.33

^1^ Italian standard population, Census 2011. The rate ratios statistically significant are highlighted in bold.

**Table 2 ijerph-19-06498-t002:** Number of deaths and age-standardized ^1^ mortality rates (ASR) from all causes and groups of causes in women at all ages in 2015–2019 and 2020, the rate ratio and the corresponding 95% confidence interval (CI). Province of Pavia, Italy.

	2015–2019		2020		2020 vs. 2015–2019		
	Average Deaths	ASR	Deaths	ASR	Rate Ratio	95% CI	
Total, all causes of death	3591	1040.1	4771	1304.4	1.25	1.21	1.30
**Group of causes**							
COVID-19	-	-	758	213.9	-	-	-
Infectious and parasitic diseases	98.4	29.3	142	38.4	**1.31**	**1.08**	**1.58**
Neoplasms	861.6	271.6	860	263.5	0.97	0.90	1.04
Malignant neoplasms of lymphoid, haematopoietic and related tissue	20.4	5.9	24	6.0	1.02	0.65	1.60
Endocrinuous, nutritional and metabolic diseases	128.2	37.0	150	40.6	1.10	0.92	1.31
Diabetes mellitus	93.2	26.8	111	30.3	1.13	0.92	1.40
Mental and behavioral disorders	184.6	50.1	230	57.6	1.15	0.99	1.33
Diseases of the nervous system and sense organs	196.6	57.4	229	65.5	1.14	0.98	1.32
Dementia and Alzheimer’s	282.2	77.5	331	85.1	1.10	0.97	1.24
Cardiovascular diseases	1371.8	381.3	1419	365.0	0.96	0.90	1.01
Ischemic heart disease	312.4	88.1	288	76.9	**0.87**	**0.77**	**0.99**
Cerebrovascular disease	424.8	118.2	422	107.5	0.91	0.82	1.01
Diseases of respiratory system	268.2	74.8	366	97.5	**1.30**	**1.16**	**1.47**
Flu virus	6.4	1.8	20	5.3	**2.91**	**1.65**	**5.13**
Pneumonia	113.8	31.2	141	37.0	1.19	0.98	1.43
Chronic lower respiratory tract diseases	83.8	24.0	97	26.0	1.09	0.87	1.36
Diseases of the digestive system	137.4	40.9	132	35.3	0.86	0.72	1.05
Diseases of the skin and subcutaneous tissue	4.4	1.2	9	2.6	**2.22**	**1.01**	**4.88**
Diseases of the musculoskeletal system and connective tissue	16.2	4.7	28	8.1	**1.70**	**1.10**	**2.64**
Diseases of the genitourinary system	83.2	23.6	92	24.0	1.02	0.81	1.28
Complications of pregnancy, childbirth and the puerperium	0	-	0	-	-	-	-
Certain conditions originating in the perinatal period	2.6	1.0	3	1.2	1.20	0.34	4.25
Congenital malformations, deformations and chromosomal abnormalities	2.4	0.9	4	1.3	1.53	0.48	4.91
Symptoms, signs and abnormal clinical and laboratory findings, not elsewhere classified	100.8	27.7	182	44.3	**1.60**	**1.35**	**1.90**
External causes of mortality	114.2	32.8	143	39.7	**1.21**	**1.01**	**1.46**
Transport accidents	6.4	2.2	9	3.2	1.43	0.68	3.0
Accidental falls	36.6	10.3	24	6.7	0.65	0.43	1.01
Suicide and intentional self-harm	9.6	3.3	9	3.3	1.0	0.49	2.03

^1^ Italian standard population, Census 2011. The rate ratios statistically significant are highlighted in bold.

**Table 3 ijerph-19-06498-t003:** Age-standardized ^1^ mortality rates (ASR) per 100,000 from COVID-19 diseases, neoplasms, and cardiovascular diseases, in 2015–2019 and 2020, number of deaths in 2020 and the rate ratio (2020 vs. 2015–2019) with the corresponding 95% confidence interval (CI), according to sex and age groups. Province of Pavia, Italy.

	Men			Women		
	COVID-19	Neoplasms	Cardiovascular Diseases	COVID-19	Neoplasms	Cardiovascular Diseases
**50–64 years old**						
ASR 2015–2019	-	259.2	122.4	-	206.4	40.4
ASR 2020	111.1	199.8	113.5	53.9	177.6	42.8
Deaths 2020	66	120	68	32	109	26
Rate ratio (95% CI)		**0.77 (0.65–0.92)**	0.93 (0.72–10.2)		0.86 (0.71–1.05)	1.06 (0.69–1.63)
**65–79 years old**						
ASR 2015–2019	-	1119.4	603	-	595.4	298.2
ASR 2020	792.8	952.2	604	313.1	580.4	300.7
Deaths 2020	318	385	242	146	272	140
Rate ratio (95% CI)		**0.85 (0.77–0.94)**	1.00 (0.84–1.19)		0.98 (0.86–1.11)	1.01 (0.83–1.22)
**≥80 years old**						
ASR 2015–2019	-	3136.8	5086.2	-	1486.7	3888.1
ASR 2020	3137	2931.2	4224.4	1808.3	1445.1	3673.8
Deaths 2020	468	441	587	576	446	1247
Rate ratio (95% CI)		0.93 (0.84–1.04)	**0.83 (0.76–0.91)**		0.97 (0.88–1.08)	0.95 (0.89–1.00)

^1^ Italian standard population, Census 2011. The rate ratios statistically significant are highlighted in bold.

## Data Availability

Not applicable.

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
