# Peer review of "Impact of the COVID-19 Pandemic on Total and Cause-Specific Mortality in Pavia, Northern Italy"

_ijerph, 2022, doi:10.3390/ijerph19116498_

Round 1

Reviewer 1 Report

 Some minor rephrasing to this sentence in the Conclusions “Our findings are partially consistent with previous reports, in particular the excess mortality in Pavia was than Italy but lower than Lombardy. “ (higher than Italy..)

Author Response

We thank the reviewer for the comment. We corrected the sentence in the Conclusion.

Reviewer 2 Report

In this manuscript Perotti and colleagues reported the impact of the pandemic on the overall and cause-specific mortality in one province in northern Italy, Pavia.

The main strengths of this paper is that it addresses an interesting and timely question, offers potential explanations based on a carefully selected set of rules, and provides clear answers. The study was conducted in acceptable way and the outcomes were accurately reported.

The authors highlighted that COVID-19 was the third cause of death in Pavia. Interestingly they assume that rises in mortality from causes related to infectious and respiratory diseases could be possibly explained due to under-diagnosis in deaths certification from COVID-19. On the other hand, the reduction in deaths from cardiovascular causes and/or neoplasms could be attributed to the harvesting effect.

The study is meticulously performed in terms of the thorough statistical analyses.

Overall, this is a well written article by experts on the field providing useful and novel information about cause-specific excess mortality during COVID-19 pandemic in one province in northern Italy, Pavia.

Minor comments 

1) Table 3 is hard to read. Please make all appropriate changes. 

2) In tables 1 and 2 please change the term << ischemic diseases>>, because the Reader will not understand which diseases are classified under this term. 

Author Response

We thank the reviewer for the comments. Here below the point-by point replies:

1) Table 3 is hard to read. Please make all appropriate changes. 

RE: We revised the Table 3, exchanging the rows and columns in order to make it more readable in the portrait page.

2) In tables 1 and 2 please change the term << ischemic diseases>>, because the Reader will not understand which diseases are classified under this term. 

RE: We edited the term “ischemic diseases” as “ischemic heart diseases” in Table 1 and 2.

Reviewer 3 Report

Dear Author

The article well written and having average scientific soundness as per current scenario. please mention the level of significance during statistical validation of findings.

Results should be written in past tense. Therefore, need to check tense and spelling error wherever applicable across the article.

Need to improve the Fig S1 with more predictable graphic patterns.

Page No. 2 line No. 55-60 in introduction section provided with certain data, but not supported by suitable citations. Could you please explain why???

You have quoted "(direct method, Italian standard pop- 75
ulation, 2011)" Page 2 line No. 75-81. But not cited reference as per standard format. Please explain???

Page no. 6 Line No. 152-170, data discussed in details but not compared with any published literature. Please find the suitable literature as per suggestion from the said literature and cute them with suitable one for quality improvement and dat validation and scientific soundness as well.

Author Response

We thank the reviewer for the comments. Here below the point-by-point replies:

The article well written and having average scientific soundness as per current scenario. please mention the level of significance during statistical validation of findings.

RE: We reported the rate ratios with their confidence intervals to validate the comparison between rates in 2020 and 2015-2019.

Results should be written in past tense. Therefore, need to check tense and spelling error wherever applicable across the article.

RE: We described the results in past tense.

Need to improve the Fig S1 with more predictable graphic patterns.

RE: We improved the Figure S1 drawing dashed line for the average number of deaths 2015-2019 and solid line for deaths in 2020.

Page No. 2 line No. 55-60 in introduction section provided with certain data, but not supported by suitable citations. Could you please explain why???

RE: We reported internal data registered by the Epidemiological Observatory in the Health Protection Agency. We added a sentence in the Introduction to explain it. In addition, we added a reference, related to the COVID-19 data.

You have quoted "(direct method, Italian standard population, 2011)" Page 2 line No. 75-81. But not cited reference as per standard format. Please explain???

RE: We added the reference to the Italian population, Census 2011, used as standard.

Page no. 6 Line No. 152-170, data discussed in details but not compared with any published literature. Please find the suitable literature as per suggestion from the said literature and cute them with suitable one for quality improvement and dat validation and scientific soundness as well.

RE: We added some references though the Discussion.
